# The concurrent impacts of drought and leaf harvesting on two traditional African vegetable non-timber forest product species

**Gisele K. Sinasson S.** *, **Charlie M. Shackleton**

Department of Environmental Science, Rhodes University, Makhanda, South Africa

* sinasson.gisele@gmail.com

**Data Availability Statement:** All data files are available from the Dryad database (https://doi.org/10.5061/dryad.z08kprrgv).

## Abstract

Investigating the concurrent impacts with harvesting on wild vegetables can guide their sustainable management while contributing to the understanding of such impacts on NTFP species. This study investigated leaf production, morphological and growth responses to the concurrent impacts of drought and leaf harvesting between two wild vegetables. A randomized greenhouse experiment was implemented with 1,334 plants of *Amaranthus sp*. and 391 of *B. pilosa*. A drought treatment was first implemented through six levels of drought stress and a control treatment. The harvesting treatment consisted of four harvesting levels and was implemented twice. Measurements were recorded before first and second harvests and at end of experiment. Data were separated into two periods (after first and second harvests) and analyzed using Multivariate Analysis of Variance and log-linear analysis. The results showed significant effects of drought on both species. However, *Amaranthus sp*. appeared more resilient to reduction in the daily amount of water than reduction in the frequency, while *B. pilosa* was resilient under both facets of drought stress. For *Amaranthus sp*., basal diameter, its growth, leaf production and survival increased with increase in the harvesting level (with some exceptions) after first harvest. After second harvest, there was decrease in plant height and leaf production. In *B. pilosa*, the impact was only significant on survival and leaf production (after first harvest). The effect of the interaction of the two drivers was significant for *Amaranthus sp*., but not for *B. pilosa*. The results also highlighted the possible negative impact of a prolonged high rate harvesting on the species performance, especially under severe drought. Basal diameter, its growth, survival and leaf production appeared more resilient to reduced amounts of watering in *Amaranthus sp*., and under both types of drought stress for *B. pilosa*. This suggests that both species could be sustained under medium drought stress.

## Introduction

The harvest of Non-Timber Forest Products (NTFPs) is widely acknowledged to affect many plant species, though the extent of effect depends on the result of interacting social, economic, and biological variables [1–3]. The impacts of NTFP harvesting are experienced not only at the plant scale but also at the population, community and ecosystem levels [2, 4–7]. However, the

**Funding:** This work was supported by the South African Research Chairs Initiative of the Department of Science and Technology and the National Research Foundation of South Africa (grant no. 84379) hold by CMS (https://www.ru.ac.za/researchgateway/researchfoci/sarchi/). Any opinion, finding, conclusion or recommendation expressed in this material is that of the authors The funders had no role in study design, data collection and analysis, decision to publish, or preparation of the manuscript.

**Competing interests:** The authors have declared that no competing interests exist.

harvest of different plant parts is not the only stress that NTFP species face because most are also subject to multiple other, often simultaneous, pressures such as competition, land use change, habitat disturbance, fire, invasive species, drought, herbivory and pests and diseases. These stressors are known to individually alter population viability and the persistence of numerous species [8–10]. The concurrent effects of two or more stressors on plant populations can be more severe than one stressor alone. For example, the interactions between harvesting and invasive species on the population viability of *Boswellia serrata* subjected to gum-resin harvesting [11], or between fire and fruit harvests from *Phyllanthus* spp. and *Terminalia chebula* [12]. However, this will depend on different species and socio-ecological contexts, and consequently population management may require adaptive guidelines. However, most studies to determine the sustainability of NTFP harvesting regimes typically account for only a single stressor, namely harvesting, and ignore how the effects of harvesting might be influenced by concurrent effects of other stressors. This means that such studies are likely to overestimate the permissible harvesting levels because potential concurrent and interactive stressors have not been accounted for. A great deal more work is required in this regard [13, 14] to help establish ecological responses and ultimately more realistic harvesting guidelines [11, 15, 16], especially for species with high intensity use, such as traditional African vegetable species.

Traditional food systems have long used a diverse range of plants harvested from the wild, including fruit-trees, edible climbing plants, nuts, tubers, and leafy greens [17–19]. Leafy greens, also termed Traditional African vegetables (TAVs [20]), are composed of important NTFP species that are relished worldwide amongst rural and urban communities for their leaves and young shoots. TAVs are globally recognized as vital sources of nutrients and their integration in alimentation can help improve household food security and dietary diversity [21, 22]. Indeed, many TAVs are reported to be rich in diverse nutrients (including calcium, iron, magnesium, phosphorus, potassium, and zinc) and in antioxidants, as compared to commonly marketed conventional vegetables [23], although more research is needed to better understand the variation in nutrient bioavailability in different species, postharvest and processing systems [24–26]. Furthermore, many of these TAVs are used medicinally to treat multiple diseases and diverse infections [27]. Thus, it has been proposed by numerous authors that TAVs should be given greater consideration in terms of food security in Africa [28–30], not only because they are highly nutritious, but also because some are deemed to be better adapted to local conditions and stressors, such as pests and drought [31, 32]. Such important TAVs in South Africa include *Amaranthus sp.* (Amaranthaceae) and *Bidens pilosa* L. (Asteraceae) and are mainly harvested from the wild [33–35]. Therefore, understanding the nature of the concurrent impacts of harvesting with other stressors, such as drought, is important to guide the sustainable use and management of these species, while giving some insights into the concurrent impacts with harvesting on NTFP species in general.

Drought stress is a key threat to the growth, production and survival of many plant species through changes in soil moisture and nutrient composition, although this impact is species-dependent and varies according to environmental conditions [36, 37]. Furthermore, the effects of drought might exacerbate the impacts of harvesting on some NTFP species, which needs to be better understood to ascertain generalizable patterns [13]. This is particularly so in light of climate change which will have marked effects on the nature, timing and amount of precipitation across much of sub-Saharan Africa, with an intensification of the frequency and intensity of droughts. However, there is still limited research on the effects of drought stress on TAVs [38–40]. Also, drought is experienced not just in terms of the reduction in the amount of rainfall but also as a decrease in the frequency of rainfall, and this variation in the expression of drought should be integrated into the investigation on the drought tolerance of wild and cultivated species. Yet, the few studies on the impact of drought on wild vegetables in southern

Africa have focused on the reduction in the amount of water received, while the effects of lower frequency of watering has rarely been investigated [38, 41]. Findings from these studies showed some TAVs to be drought tolerant, though the level of tolerance is species dependent and varies according to different plant characteristics such as morphology, growth, physiology and production. Furthermore, Zhang *et al*. [36] indicated that a decrease in the frequency of rainfall could have more severe effects on plant growth and survival than the reduction in the amount. Therefore, it is important to understand how these plants respond to the two different aspects of drought stress (i.e. reduction in the amount and frequency of watering). Moreover, these studies did not examine the interactive effects of drought tolerance with harvesting.

Thus, this study used a greenhouse experiment to assess the concurrent effects of drought and leaf harvesting on *Amaranthus sp*. (green morphotype) and *Bidens pilosa*, using different levels of leaf harvesting and drought stress (incl. reduction in the amount and in the frequency of watering). The key questions addressed were: (i) how strong are the concurrent impacts of drought with leaf harvesting on *Amaranthus sp*. and *Bidens pilosa*? (ii) What are the plant responses (i.e. survival, morphology, growth and leaf production) to the concurrent impacts of the two stressors? (iii) Are the concurrent impacts of the two stressors species-specific under a common environment (e.g. greenhouse)?

## Methods

### Greenhouse experiment design

The greenhouse experiment was conducted between February and September 2021, at Rhodes University (33˚18'S; 26˚32'E) located in Makhanda, in the Eastern Cape, South Africa. Maximum and minimum temperatures and relative humidity in the greenhouse were regulated by the ambient conditions. Radiation was provided by sunlight [42]. The Eastern Cape falls within the warm and temperate region, with the mean temperature ranging between 24.5–25˚C and the mean rainfall between 550–700 mm [43]. It is currently experiencing severe drought and temperature increase, although there are cases of decrease in temperature in some localities [44, 45]. The maximum temperature in Makhanda, during the experiment, ranged from 18˚C to 32˚C in February-May and from 11˚C to 32˚C in June-September, while the minimum values ranged from 6˚C to 23˚C in February-May and from 1˚C to 15˚C in June-September [46]. The water used in the experiment was from the standard municipal supply.

**Obtaining plants and plant preparation.** A preliminary seed germination assay was realized on site to produce seedlings for the experiment. For that, *Amaranthus* seeds were obtained from the garden of a grower of TAVs in Buffalo City (32˚56'S; 27˚38'E) and *Bidens* seeds were collected from the roadside near Howick (29˚30'S; 30˚13'E). For *Amaranthus sp*., seeds were first soaked in cool water for two hours to remove any seed dormancy. *B. pilosa* seeds do not require any pre-germination treatment and so they were just lightly sowed in the soil.

A seed germination assay was conducted using plastic germination trays (Fig 1A). The local topsoil used to fill the trays was collected next to the greenhouse and was clay loam. The trays were watered before sowing the seeds and regularly after sowing until the end of the assay. Seed germination was considered as the visible emergence of the radicle [47].

After germination, seedlings were watered as needed until the three-leaf stage. At the three-leaf stage, 1,337 seedlings of uniform size for *Amaranthus sp*. and 399 for *B. pilosa* were transplanted into small plastic pots (14-cm of height and 8-cm diameter; Fig 1B). The pots were filled with the same soil used for the germination assay. However, later on and before the application of the experimental treatments, a same amount of river soil was added to all the pots to lessen the drawbacks of clay loam on the plants. Plants were left for another two weeks (336 hours) before starting with experimental treatments. Pots were watered daily at 100% pot

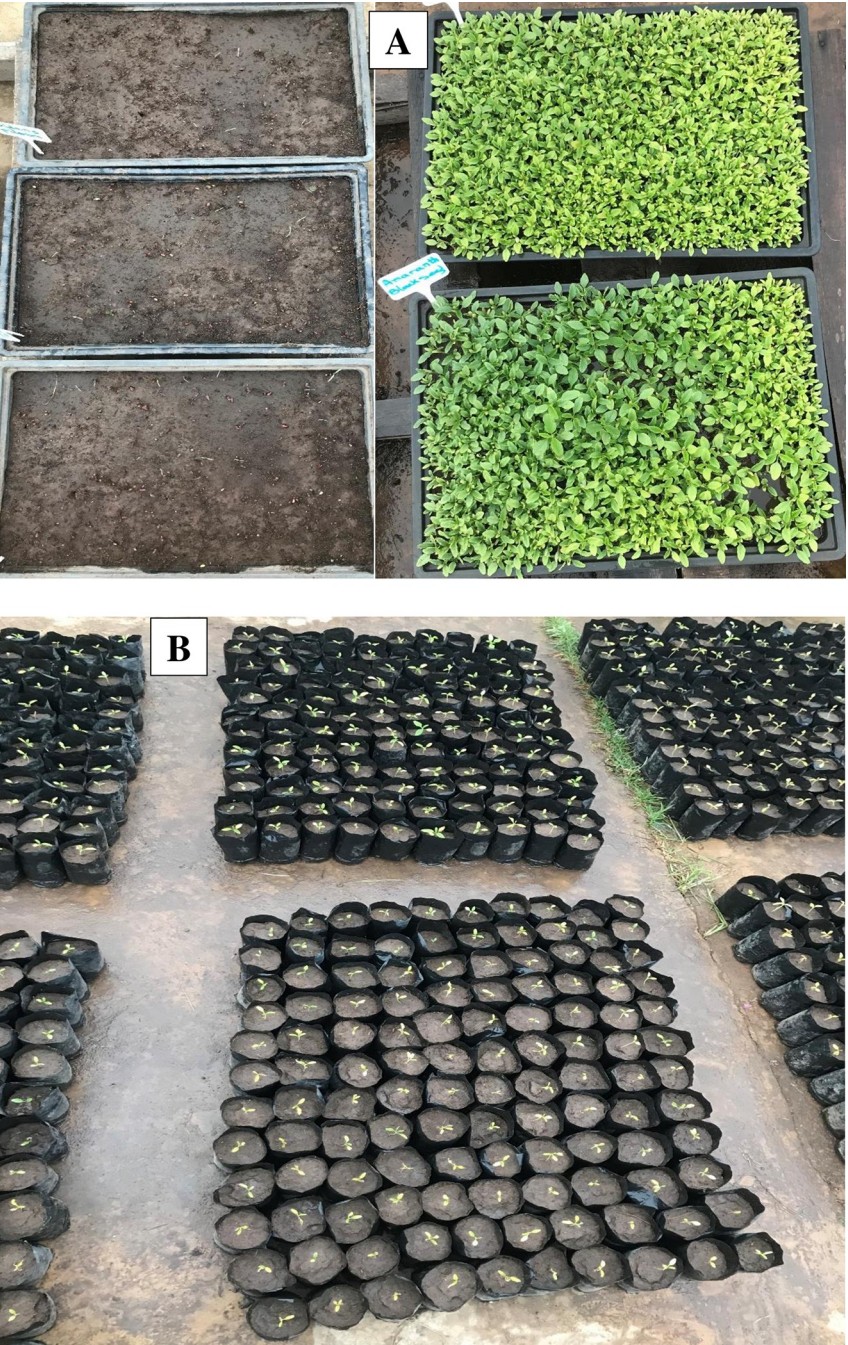

**Fig 1.** Images illustrating the germination assay in plastic trays (A) and seedlings transplanted in small pots for experimental treatments (B).

capacity (PC). The field capacity or water holding capacity (WHC) of 100 g of soil was determined by adopting the "Cylinder bath method", i.e. saturating with water and allowing drainage for two hours [48]. The PC was extrapolated from the remaining water in the soil, by weighing the soil in the pots; the mass of the plants being considered negligible. In the case of this study, the determined PC was 120 ml. Each pot was labelled to denote application of treatments and data collection.

**Experimental treatments.**   The drought treatment was first implemented through six levels of drought stress (or drought treatments) including three reduced amounts of watering and three frequencies of watering, as follows: (i) 25% pot capacity (PC) once every 24 hours (25% PC), (ii) 50% PC once every 24 hours (50% PC), (iii) 75% PC once every 24 hours (75% PC), (iv) PC once every three days (72 hours) [PC(3)], (v) PC once every six days (144 hours) [PC(6)] and (vi) PC once every nine days (216 hours) [PC(9)]. A control treatment with adequate daily water supply, i.e. PC once every 24 hours (PC) was also designed to allow comparison of the effect of drought stress on plant growth and survival, and to determine the optimal water treatment for the study species. For each species, each of the seven water regimes (or treatments) was applied to a random sample of the pots (191 replications in the case of *Amaranthus sp.* and 57 replications for *B. pilosa*) until the end of the experiment.

In terms of the harvesting treatment, this study focused on the harvesting of leaves copying harvesting practices of rural communities in the region. The harvesting treatment consisted of four levels of harvesting, represented by different percentages of leaves harvested: (i) 0% of leaves removed (0%LH), (ii) 50% of leaves removed starting with the largest leaves (50%LH), (iii) 100% of leaves removed (100%LH) and (iv) all the leaves removed with the supporting part of stem (Cut). However, the fourth harvesting level was only implemented for *Amaranthus sp*. This is because, after discussion with some local harvesters, it appeared that they only harvest the leaves of *B. pilosa*, while in the case of *Amaranthus sp*., they may also cut the young tender stems along with the leaves. Plants within each water regime were randomly divided into three or four groups (depending on the species) and each group was subjected to a level of harvesting. The harvesting treatment was repeated twice during the course of the study to evaluate the impact of prolonged harvesting on the plants.

**Monitoring plant growth, vigor and production.**   Plant morphology and growth as well as leaf production and survival were monitored during the experiment. To assess the morphology and growth, the shoot length (as an estimate of height) and basal diameter of the plants were measured at three different times as follows: (i) before first harvest (three weeks of drought treatment for *Amaranthus sp*. and five weeks for *Bidens pilosa*), before second harvest (11 weeks after first harvest for *Amaranthus sp*. and six weeks for *Bidens pilosa*) and at the end of the experiment (six weeks after second harvest for *Amaranthus sp*. and eight weeks for *Bidens pilosa*). For leaf production, the number of leaves was recorded per plant at the three above-described measurement times. For the measurements, a standard ruler was used for the height and calipers for the basal diameter. The number of dead individuals was also recorded.

## Data analysis

Mean values of the different variables considered and related to the morphology (basal diameter and height), growth (growth in diameter and height) and leaf production (number of leaves) were first determined per replicate (water treatments or harvesting levels) and within two treatment periods (after first and second harvests). The survival rates were also determined. To evaluate the individual impact of drought and harvesting treatments on the combination of the different variables considered and within each of the two treatment periods, different one-way Multivariate Analysis of Variance (MANOVAs) were performed. The impact of the interaction of the two treatments was analyzed, using two-way MANOVAs [49]. To assess whether or not the leaf production, morphological and growth responses to the different treatments were species dependent, species was included as a factor into two-way (for individual impacts) and three-way (for the interaction of the two treatments) MANOVAs [49]. The impact of drought and leaf harvesting treatments on the survival of each species was evaluated, using log-linear models. In the case of significant MANOVAs, the results of the

Analysis of Variance (ANOVA) on individual dependent variables were assessed to situate the characteristics that best discriminated the treatments.

For the different MANOVAs, the Pillai's Trace test was used; it is a reasonable choice because it is considered to be more robust, especially for violations of certain test assumptions [49]. The Pillai's Trace statistics are then converted to F-statistics since they do not have test distributions. The effect size of the different impacts was evaluated using the partial eta-squared which is an estimate of the proportion of the variance in the characteristics or dependent variables (between the different replicates) that is explained by the different factors or independent variables [50]. It measures the effect the independent variable(s) has on the dependent variables. The value for the partial eta-squared ranges from 0 to 1; values below 0.14 indicate a weak impact of the factor(s) considered while values above 0.14 show a large impact.

All the data analyses were done using R 4.1.3 [51].

## Results

### Leaf production, growth and morphological responses of *Amaranthus sp.* and *B. pilosa* to drought treatment

The results from the multivariate analysis of variance (MANOVA) showed that at least one water treatment was significantly different from others based on the combination of leaf production, growth and morphological characteristics after first and second harvests, for both species ($p < 0.001$; Table 1). However, the impact of drought treatment was weak (effect size < 0.14) after harvesting for both species and it varied between the two species ($p < 0.001$). The effect size of the drought treatment was slightly weaker for *Amaranthus sp.* than for *B. pilosa* after first harvest ($p < 0.001$), while the contrary was observed after the second harvest ($p = 0.001$). Also, there were differences between the two species in terms of the individual

**Table 1. Results of the Multivariate Analysis of Variance and log-linear analysis for the impact of water treatment on the study species.**

| Characteristics | *Amaranthus sp.* | | | *Bidens pilosa* | | |
|---|---|---|---|---|---|---|
| Statistics | Effect size | F-value | P-value | Effect size | F-value | P-value |
| *After first harvest* | | | | | | |
| Model | 0.06 | 5.172 | <0.001 | 0.10 | 3.834 | <0.001 |
| Basal diameter (mm) | | 5.583 | <0.001 | | 1.766 | 0.107 |
| Total height (cm) | | 0.674 | 0.670 | | 0.373 | 0.895 |
| No. of leaves (per plant) | | 2.844 | 0.010 | | 5.049 | < 0.001 |
| Growth in diameter (mm/week) | | 4.900 | <0.001 | | 1.207 | 0.304 |
| Growth in height (cm/week) | | 3.925 | <0.001 | | 0.920 | 0.482 |
| **Water_Treat*Species** | | **2.311** | **<0.001** | | **-** | **-** |
| Survival rate (%)[1] | | - | <0.001 | | - | <0.001 |
| *After second harvest* | | | | | | |
| Model | 0.11 | 2.475 | <0.001 | 0.09 | 2.046 | < 0.001 |
| Basal diameter (mm) | | 4.836 | <0.001 | | 0.402 | 0.876 |
| Total height (cm) | | 3.073 | 0.008 | | 2.180 | 0.049 |
| No. of leaves (per plant) | | 0.842 | 0.540 | | 3.857 | 0.001 |
| Growth in diameter (mm/week) | | 5.032 | <0.001 | | 1.502 | 0.183 |
| Growth in height (cm/week) | | 2.025 | 0.067 | | 1.215 | 0.303 |
| **Water_Treat*Species** | | **2.009** | **0.001** | | **-** | **-** |
| Survival rate (%)[1] | | - | 0.002 | | - | 0.002 |

[1]: log-linear analysis

characteristics that best discriminated the water treatments. Results of the log-linear analysis indicated a significant difference between water treatments based on plant survival rate after both first ($p < 0.001$) and second ($p = 0.002$) leaf harvests, for both species (Table 1).

**Water treatment effects after first harvest.** The impact of water treatment was significant for all the characteristics considered for *Amaranthus sp.*, except height. In the case of *B. pilosa*, there were significant differences between water treatments only for the number of leaves per plant and survival rate (Table 1).

For *Amaranthus sp.*, the results highlighted a decrease in the mean basal diameter and growth in diameter with a reduction in both the daily amount and the frequency of watering, with the decrease being more pronounced with the reduction in the frequency of watering. (Table 2). A similar trend was observed for the survival rate, for reduced amount until 50%PC every day and reduced frequency PC(6). Concerning the growth in height and leaf production, the results showed an increase with a reduction in both the daily amount and the frequency of watering, except 75%PC every day and PC(3). However, the increase in the leaf production was more pronounced in the reduced daily amount than the reduced frequency of watering.

For *B. pilosa*, the results showed an increase in the number of leaves per plant and the survival rate with a reduction in both the daily amount (25%PC every day) and the frequency (PC (6) and PC(9)) of watering (Table 2). However, the increase in the leaf production was higher with the reduction in the frequency of watering, while the increase in the survival rate was more pronounced with the reduction in the daily amount of watering. A similar trend (i.e. an increase with a reduction in both the daily amount and the frequency of watering) was observed for basal diameter, height and growth in height, although the differences between treatments were not significant.

**Water treatment effects after second harvest.** For *Amaranthus sp.*, there were significant differences between water treatments for all the characteristics considered, except number of leaves per plant and growth in height. For *B. pilosa*, the differences were significant only for height, number of leaves per plant and survival rate (Table 1).

For *Amaranthus sp.*, the results showed an increase in basal diameter and growth in diameter with a reduction in the daily amount of watering (except 25%PC for basal diameter), while

**Table 2. Mean values of the considered characteristics (± SD) of the study species between water treatments, after first harvest.**

| Characteristics | 25%PC | 50%PC | 75%PC | PC | PC(3) | PC(6) | PC(9) |
|---|---|---|---|---|---|---|---|
| *Amaranthus sp.* | *n = 87* | *n = 72* | *n = 65* | *n = 80* | *n = 53* | *n = 64* | *n = 98* |
| Basal diameter (mm) | 2.37 ± 0.7 | 2.47 ± 0.7 | 2.52 ± 0.8 | 2.52 ± 0.7 | 2.28 ± 0.6 | 2.05 ± 0.5 | 2.18 ± 0.6 |
| Total height (cm) | 7.09 ± 2.9 | 6.78 ± 2.7 | 6.65 ± 2.9 | 6.72 ± 3.0 | 6.25 ± 2.7 | 6.72 ± 2.9 | 7.04 ± 3.5 |
| No. of leaves (per plant) | 10 ± 7 | 8 ± 7 | 7 ± 7 | 6 ± 6 | 6 ± 6 | 7 ± 9 | 8 ± 9 |
| Growth in diameter (mm/week) | 0.03 ± 0.0 | 0.04 ± 0.0 | 0.05 ± 0.0 | 0.06 ± 0.1 | 0.02 ± 0.0 | 0.04 ± 0.1 | 0.03 ± 0.1 |
| Growth in height (cm/week) | 0.10 ± 0.2 | 0.03 ± 0.3 | -0.08 ± 0.3 | 0.01 ± 0.4 | -0.04 ± 0.3 | 0.10 ± 0.4 | 0.09 ± 0.3 |
| Survival rate (%) | 45.8 | 37.7 | 34.0 | 41.9 | 28.0 | 33.7 | 51.3 |
| *Bidens pilosa* | *n = 46* | *n = 32* | *n = 34* | *n = 31* | *n = 29* | *n = 37* | *n = 45* |
| Basal diameter (mm) | 2.14 ± 0.8 | 2.10 ± 0.7 | 2.02 ± 0.8 | 1.69 ± 0.6 | 1.74 ± 0.6 | 1.80 ± 0.7 | 1.79 ± 0.6 |
| Total height (cm) | 10.61 ± 6.8 | 9.86 ± 6.7 | 10.31 ± 7.5 | 8.35 ± 6.0 | 9.07 ± 7.7 | 10.68 ± 7.2 | 11.17 ± 7.5 |
| No. of leaves (per plant) | 12 ± 7 | 8 ± 6 | 8 ± 5 | 9 ± 6 | 7 ± 3 | 10 ± 7 | 14 ± 8 |
| Growth in diameter (mm/week) | 0.07 ± 0.1 | 0.08 ± 0.1 | 0.09 ± 0.1 | 0.08 ± 0.1 | 0.09 ± 0.1 | 0.09 ± 0.1 | 0.09 ± 0.1 |
| Growth in height (cm/week) | 0.64 ± 0.9 | 0.50 ± 0.8 | 0.45 ± 0.8 | 0.21 ± 0.5 | 0.23 ± 0.8 | 0.40 ± 0.8 | 0.58 ± 1.0 |
| Survival rate (%) | 82.1 | 58.2 | 60.7 | 56.4 | 50.9 | 66.1 | 80.4 |

PC = Pot capacity; (3), (6), (9) = every three, six and nine days, respectively; SD = standard deviation; n = sample size.

**Table 3. Mean values of the considered characteristics (± SD) of the study species between water treatments, after second harvest.**

| Characteristics | 25%PC | 50%PC | 75%PC | PC | PC(3) | PC(6) | PC(9) |
|---|---|---|---|---|---|---|---|
| *Amaranthus sp.* | *n = 24* | *n = 20* | *n = 19* | *n = 29* | *n = 16* | *n = 6* | *n = 15* |
| Basal diameter (mm) | 2.99 ± 0.6 | 3.08 ± 0.5 | 3.26 ± 0.8 | 2.99± 0.5 | 2.53 ± 0.4 | 2.40 ± 0.8 | 2.46 ± 0.6 |
| Total height (cm) | 4.86 ± 3.4 | 4.89 ± 2.4 | 4.26± 2.1 | 4.79 ± 1.8 | 3.49 ± 0.6 | 8.85 ± 4.2 | 4.87 ± 4.0 |
| No. of leaves (per plant) | 3 ± 4 | 3 ± 4 | 2 ± 4 | 3 ± 4 | 1 ± 1 | 3 ± 5 | 4 ± 8 |
| Growth in diameter (mm/week) | 0.01 ± 0.1 | 0.03 ± 0.0 | 0.03 ± 0.1 | 0.00 ± 0.1 | -0.06 ± 0.1 | -0.11 ± 0.1 | -0.11 ± 0.1 |
| Growth in height (cm/week) | -0.28 ± 0.2 | -0.23 ± 0.2 | -0.15 ± 0.3 | -0.44 ± 0.4 | -0.30 ± 0.2 | -0.31 ± 0.3 | -0.22 ± 0.3 |
| Survival rate (%) | 12.6 | 10.5 | 9.9 | 15.2 | 8.5 | 3.2 | 7.9 |
| *Bidens pilosa* | *n = 33* | *n = 21* | *n = 25* | *n = 17* | *n = 13* | *n = 23* | *n = 28* |
| Basal diameter (mm) | 2.38 ± 0.7 | 2.23 ± 0.7 | 2.33 ± 0.8 | 2.14 ± 0.6 | 2.46 ± 0.4 | 2.28 ± 0.5 | 2.29 ± 0.5 |
| Total height (cm) | 15.52 ± 6.1 | 12.42 ± 6.6 | 15.25 ± 6.0 | 11.68 ± 6.8 | 15.51 ± 6.2 | 16.77 ± 5.1 | 15.12 ± 4.9 |
| No. of leaves (per plant) | 13 ± 10 | 9 ± 7 | 12 ± 8 | 11 ± 8 | 13 ± 8 | 19 ± 10 | 20 ± 9 |
| Growth in diameter (mm/week) | 0.00 ± 0.1 | -0.04 ± 0.1 | 0.00 ± 0.1 | 0.00 ± 0.1 | 0.03 ± 0.1 | 0.02 ± 0.1 | 0.02 ± 0.1 |
| Growth in height (cm/week) | 0.01 ± 0.5 | -0.05 ± 0.7 | 0.28 ± 0.8 | 0.15 ± 0.9 | 0.27 ± 0.8 | 0.44 ± 1.0 | 0.12 ± 1.0 |
| Survival rate (%) | 58.9 | 38.2 | 44.6 | 30.9 | 22.8 | 41.1 | 50.0 |

PC = Pot capacity; (3), (6), (9) = every three, six and nine days, respectively; SD = standard deviation; n = sample size.

there was a decrease with a reduction in the frequency of watering (Table 3). Concerning the height, results indicated an increase with a reduction in both the daily amount and the frequency of watering except 75%PC and PC(3). In terms of plant survival, a more pronounced decrease in the rate was observed with the reduction in the frequency of watering than with the reduction in the daily amount of watering.

For *B. pilosa*, there was an increase in plant height and number of leaves per plant with a reduction in both the daily amount and the frequency of watering (except 50%PC for leaf production; Table 3). However, the increase in the leaf production was higher with the reduced frequencies of watering. A similar trend was observed for plant survival except PC(3), although the increase was more pronounced with the reduced amount of watering 25%PC than the reduced frequencies of watering PC(6) and PC(9). An increase in basal diameter was observed with the reduced daily amounts and frequencies of watering, although the differences between treatments were not significant.

### Leaf production, growth and morphological responses to the impact of leaf harvesting between *Amaranthus sp*. and *Bidens pilosa*

The results from the MANOVA showed a significant impact of leaf harvesting on the combination of leaf production, growth and morphological characteristics after both the first and second harvests, only for *Amaranthus sp.* ($p < 0.001$; Table 4); the impact on *B. pilosa* was not significant ($p = 0.123$). This difference in the impact of leaf harvesting between the two species was significant ($p < 0.05$). The effect size of the impact on *Amaranthus sp.* was large ($> 0.14$) but was weak for *B. pilosa* ($< 0.14$). Results of the log-linear analysis showed a significant impact of leaf harvesting on plant survival after the first and second harvests, for both species (Table 4).

**Effects of leaf harvesting after first harvest.** The impact of leaf harvesting was significant for all the characteristics for *Amaranthus sp*. In the case of *B. pilosa*, although the impact was not significant on the combination of leaf production, morphological and growth characteristics, the harvesting treatments were significantly separated by the number of leaves per plant (Table 4). Results also showed a significant impact on plant survival.

**Table 4. Results of the Multivariate Analysis of Variance and log-linear analysis for the impact of leaf harvesting on the study species.**

| Characteristics | Amaranthus sp. | | | Bidens pilosa | | |
|---|---|---|---|---|---|---|
| Statistics | Effect size | F-value | P-value | Effect size | F-value | P-value |
| *After first harvest* | | | | | | |
| Model | 0.20 | 25.694 | <0.001 | 0.03 | 1.540 | 0.123 |
| Basal diameter (mm) | | 34.139 | <0.001 | | 0.098 | 0.907 |
| Total height (cm) | | 83.894 | <0.001 | | 0.165 | 0.848 |
| No. of leaves (per plant) | | 3.091 | 0.030 | | 5.594 | 0.004 |
| Growth in diameter (mm/week) | | 19.171 | <0.001 | | 1.081 | 0.341 |
| Growth in height (cm/week) | | 85.1 | <0.001 | | 0.058 | 0.944 |
| **Harvesting_Treat*Species** | | **2.743** | **0.002** | | - | - |
| Survival rate (%)[1] | | - | <0.001 | | - | 0.011 |
| *After second harvest* | | | | | | |
| Model | 0.26 | 8.623 | <0.001 | 0.05 | 1.194 | 0.295 |
| Basal diameter (mm) | | 1.210 | 0.309 | | 0.153 | 0.858 |
| Total height (cm) | | 67.325 | <0.001 | | 1.903 | 0.153 |
| No. of leaves (per plant) | | 5.040 | 0.003 | | 0.254 | 0.776 |
| Growth in diameter (mm/week) | | 1.631 | 0.186 | | 0.752 | 0.473 |
| Growth in height (cm/week) | | 1.466 | 0.227 | | 2.189 | 0.116 |
| **Harvesting_Treat*Species** | | **1.869** | **0.047** | | - | - |
| Survival rate (%)[1] | | - | <0.001 | | - | 0.010 |

[1]: log-linear analysis

For *Amaranthus sp*., the results showed an increase in the mean basal diameter, number of leaves per plant and growth in diameter with increase in the harvesting level except the treatment 50%LH which had similar or lower mean values that the unharvested treatment (Table 5). The most harvested plants displayed the lowest mean height and growth in height, compared to those harvested less. However, the treatment 50%LH had the highest mean values for these two parameters than the unharvested treatment. In terms of plant survival, there was a decrease in the rate with an increase in the harvesting level except the treatment Cut which had a highest rate than the unharvested treatment.

For *B. pilosa*, the results indicated a decrease in the mean number of leaves per plant and survival rate with an increase in the harvesting level except for the treatment 50%LH, which had similar a mean number of leaves to the unharvested treatment (Table 5). A similar trend was also observed for basal diameter and height as well as growth in diameter and height, although the differences between the treatments were not significant.

**Effects of leaf harvesting after second harvest.** For *Amaranthus sp*., there were significant differences between the harvesting treatments only for the height, number of leaves per plant and the survival rate, while for *B. pilosa*, the differences between the harvesting treatments were significant only for the survival rate (Table 4).

For *Amaranthus sp*., plant mean height and number of leaves per plant decreased with an increase in the harvesting level, except for the treatment 50%LH, which had higher mean value for height than the unharvested treatment (Table 6). In terms of plant survival, there was an increase in the rate with increase in the harvesting level except the treatment 50%LH which had similar rate with the unharvested treatment. Although the differences between the treatments were not significant, there was an increase in the mean basal diameter with an increase in the harvesting level, while results showed a more pronounced decrease in the growth in diameter and height in the unharvested treatment compared to the harvested ones.

**Table 5. Mean values of the considered characteristics (± SD) of the study species between harvesting treatments, after first harvest.**

| Characteristics | 0%LH | 50%LH | 100%LH | Cut |
|---|---|---|---|---|
| *Amaranthus sp.* | *n = 147* | *n = 118* | *n = 96* | *n = 154* |
| Basal diameter (mm) | 2.19 ± 0.6 | 2.09 ± 0.6 | 2.21 ± 0.6 | 2.75 ± 0.6 |
| Total height (cm) | 7.96 ± 2.5 | 8.10 ± 2.8 | 7.49 ± 3.0 | 4.20 ± 1.5 |
| No. of leaves (per plant) | 7 ± 6 | 6 ± 6 | 9 ± 10 | 8 ± 8 |
| Growth in diameter (mm/week) | 0.03 ± 0.1 | 0.02 ± 0.1 | 0.04 ± 0.1 | 0.06 ± 0.0 |
| Growth in height (cm/week) | 0.16 ± 0.3 | 0.18 ± 0.3 | 0.11 ± 0.3 | -0.24 ± 0.2 |
| Survival rate (%) | 47.3 | 38.2 | 31.0 | 49.7 |
| *Bidens pilosa* | *n = 89* | *n = 82* | *n = 81* | |
| Basal diameter (mm) | 1.99 ± 0.8 | 1.93 ± 0.7 | 1.78 ± 0.7 | |
| Total height (cm) | 11.10 ± 7.3 | 9.71 ± 7.0 | 9.53 ± 6.8 | |
| No. of leaves (per plant) | 11 ± 6 | 11 ± 7 | 8 ± 6 | |
| Growth in diameter (mm/week) | 0.09 ± 0.1 | 0.09 ± 0.1 | 0.06 ± 0.1 | |
| Growth in height (cm/week) | 0.53 ± 0.9 | 0.43 ± 0.8 | 0.39 ± 0.7 | |
| Survival rate (%) | 97.8 | 89.1 | 87.1 | |

LH = Leaves harvested; Cut = all the leaves harvested with the supporting part of stem; SD = standard deviation; n = sample size.

For *B. pilosa*, plant survival decreased with an increase in the harvesting rate (Table 6). A similar trend was observed for mean basal diameter and height, as well as growth in diameter and height, although the differences between the treatments were not significant. Conversely, there was an increase in the mean number of leaves per plant with an increase in the harvesting rate, except the treatment 50%LH that had a similar mean value to the unharvested treatment.

**Leaf production, growth and morphological responses to the interaction of drought and leaf harvesting between *Amaranthus sp*. and *Bidens Pilosa*.** The results from the MANOVA showed a significant and medium (effect size 0.16) interaction of water and harvesting treatments on the combination of leaf production, growth and morphological characteristics only for *Amaranthus sp*. after the second harvest ($p < 0.001$; Table 7). The interaction of the

**Table 6. Mean values of the considered characteristics (± SD) of the study species between harvesting treatments, after second harvest.**

| Characteristics | 0%LH | 50%LH | 100%LH | Cut |
|---|---|---|---|---|
| *Amaranthus sp.* | *n = 6* | *n = 6* | *n = 21* | *n = 95* |
| Basal diameter (mm) | 2.66 ± 0.6 | 2.76 ± 0.3 | 2.75 ± 0.6 | 2.96 ± 0.7 |
| Total height (cm) | 8.55 ± 3.6 | 9.03 ± 3.4 | 8.03 ± 3.4 | 3.57 ± 0.8 |
| No. of leaves (per plant) | 7 ± 7 | 7 ± 5 | 3 ± 3 | 2 ± 4 |
| Growth in diameter (mm/week) | -0.10 ± 0.1 | 0.00 ± 0.0 | 0.00 ± 0.1 | -0.01 ± 0.1 |
| Growth in height (cm/week) | -0.53 ± 0.7 | -0.18 ± 0.2 | -0.27 ± 0.4 | -0.28 ± 0.3 |
| Survival rate (%) | 1.9 | 1.9 | 6.8 | 30.6 |
| *Bidens pilosa* | *n = 62* | *n = 55* | *n = 43* | |
| Basal diameter (mm) | 2.39 ± 0.7 | 2.37 ± 0.6 | 2.10 ± 0.6 | |
| Total height (cm) | 15.35 ± 5.9 | 15.39 ± 5.4 | 11.29 ± 6.2 | |
| No. of leaves (per plant) | 14 ± 9 | 14 ± 9 | 16 ± 11 | |
| Growth in diameter (mm/week) | 0.01 ± 0.1 | 0.02 ± 0.1 | -0.02 ± 0.1 | |
| Growth in height (cm/week) | 0.31 ± 0.8 | 0.41 ± 0.8 | -0.28 ± 0.6 | |
| Survival rate (%) | 68.1 | 59.8 | 46.2 | |

LH = Leaves harvested; Cut = all the leaves harvested with the supporting part of stem; SD = standard deviation; n = sample size.

**Table 7. Results of the Multivariate Analysis of Variance and log-linear analysis for the impact of the interaction of the treatments on the study species.**

| Characteristics | Amaranthus sp. | | | Bidens pilosa | | |
|---|---|---|---|---|---|---|
| Statistics | Effect size | F-value | P-value | Effect size | F-value | P-value |
| *After first harvest* | | | | | | |
| Model | 0.04 | 1.189 | 0.112 | 0.06 | 0.997 | 0.484 |
| Basal diameter (mm) | | 0.498 | 0.959 | | 1.556 | 0.107 |
| Total height (cm) | | 1.123 | 0.326 | | 0.858 | 0.591 |
| No. of leaves (per plant) | | 1.929 | 0.012 | | 1.047 | 0.407 |
| Growth in diameter (mm/week) | | 0.711 | 0.801 | | 1.012 | 0.439 |
| Growth in height (cm/week) | | 1.175 | 0.277 | | 1.232 | 0.263 |
| **Interaction*Species** | | **1.292** | **0.066** | | - | - |
| Survival rate (%)[1] | | - | <0.001 | | - | 0.787 |
| *After second harvest* | | | | | | |
| Model | 0.16 | 2.259 | <0.001 | 0.09 | 1.126 | 0.263 |
| Basal diameter (mm) | | 1.364 | 0.214 | | 0.887 | 0.547 |
| Total height (cm) | | 8.516 | <0.001 | | 0.587 | 0.821 |
| No. of leaves (per plant) | | 2.125 | 0.033 | | 1.141 | 0.339 |
| Growth in diameter (mm/week) | | 2.124 | 0.033 | | 0.733 | 0.691 |
| Growth in height (cm/week) | | 2.198 | 0.027 | | 0.822 | 0.608 |
| **Interaction*Species** | | **0.401** | **0.946** | | - | - |
| Survival rate (%)[1] | | - | <0.001 | | - | 0.214 |

Interaction = interaction of water and leaf harvesting treatments; [1]: log-linear analysis

two treatments was not significant on *Bidens pilosa*. The difference in the impact of the interaction of water and harvesting treatments between the two species after the second harvest was not significant ($p = 0.484$). Also, there is a significant interactive impact of water and harvesting treatments on plant survival only for *Amaranthus sp*. after both harvests ($p < 0.001$).

Although, the multivariate interaction was not significant on the combination of all the considered characteristics after the first harvest for *Amaranthus sp*., the different groups were significantly separated by the number of leaves per plant (Table 7). For the significant multivariate interaction (i.e. after second harvest), the groups were best separated by all the plant characteristics considered except basal diameter.

After the first harvest (in *Amaranthus sp*.), in the control treatment (PC) and the treatments with reduced daily amount of watering (25%, 50% and 75% PC), the results showed a decrease in the survival rate with increased harvesting, except for the treatment Cut which had higher rate than the unharvested treatment (Table 8). A similar trend was observed with the least reduced frequency of watering PC(3), although the decrease was higher in the treatment 50% LH than the treatment 100%LH. On the other hand, for the medium (PC(6) and high (PC(9) reduction in the frequency of watering, there was also a decrease in the survival rate with harvesting, without a clear pattern as to whether or not the decrease is higher with an increased harvesting level. In terms of leaf production, there was no clear trend with regards to the effect of the interaction of drought and leaf harvesting (Fig 2A). However, with the higher reductions in the amounts of watering (25%PC, 50%PC) and least reduced frequency PC(3), the treatment Cut showed the highest number of leaves per plant. With the least reduced amount of watering (75%PC) and medium reduction in frequency, the unharvested treatment displayed the highest number of leaves per plant, while the treatment 100%LH had the highest number in the control and with the high reduction in the frequency of watering.

**Table 8. Survival rates (%) of *Amaranthus sp.* in relation to water and harvesting treatments.**

| Harvesting | 25%PC | 50%PC | 75%PC | PC | PC(3) | PC(6) | PC(9) |
|---|---|---|---|---|---|---|---|
| *After first harvest* | | | | | | | |
| 0%LH | 53.3 | 41.3 | 36.4 | 52.4 | 34.1 | 48.9 | 64.4 |
| 50%LH | 40.0 | 39.1 | 34.1 | 41.9 | 16.3 | 37.2 | 57.8 |
| 100%LH | 40.0 | 23.9 | 20.5 | 19.0 | 25.0 | 45.2 | 42.6 |
| Cut | 60.0 | 52.2 | 56.8 | 71.4 | 46.3 | 15.2 | 47.8 |
| *After second harvest* | | | | | | | |
| 0%LH | 0.0 | 2.2 | 0.0 | 9.5 | 0.0 | 0.0 | 2.2 |
| 50%LH | 0.0 | 0.0 | 4.5 | 2.3 | 0.0 | 2.3 | 4.4 |
| 100%LH | 8.9 | 13.0 | 4.5 | 11.9 | 2.3 | 7.1 | 0.0 |
| Cut | 44.4 | 28.3 | 34.1 | 45.2 | 34.1 | 4.3 | 26.1 |

PC = Pot capacity; (3), (6), (9) = every three, six and nine days, respectively; LH = Leaves harvested; Cut = all the leaves harvested with the supporting part of stem.

After the second harvest (in *Amaranthus sp.*), a similar trend to the period after the first harvest was observed for plant survival in the control treatment and the treatments with reduced daily amount and least reduced frequency of watering PC(3); the treatment Cut displayed the highest rate (Table 8). However, contrary to the previous period, the rate increased with harvesting, although there were some exceptions with the treatment 50%LH or 100%LH depending on the case. For the other parameters, no clear trends were observed. However, in terms of plant height and leaf production (Fig 2B and 2C), the treatment 100%LH displayed the highest mean values with the greatest reduction in the amount of watering (25% and 50%PC), while the treatment 50%LH obtained the highest mean values with the least reduced amount of water (75% PC). The cut plants had the lowest height while plants with 100%LH had the highest number of leaves per plant in the control water treatment. The plants with 100%LH and 50%LH had the highest mean height respectively with medium and high reduction in the frequency of watering. For the growth in basal diameter (Fig 2D), the cut plants displayed the highest mean value in the water treatments 25%PC, 50%PC, PC and PC(3), while plants with 100%LH had the highest value in the water treatments 75%PC and PC(6) and those with 50%LH in the water treatment PC(9). Concerning the growth in height (Fig 2E), plants with 100%LH obtained the highest mean value in the water treatments 25%PC, 50%PC and PC(6), while the cut plants had the highest mean value in the water treatments 75%PC, PC, PC(3) and PC(9).

## Discussion

### Concurrent impacts of drought and leaf harvesting on *Amaranthus sp.* and *Bidens pilosa*

Many traditional African vegetables (TAVs) are widely acknowledged to be resilient to drought, although the empirical evidence is limited. However, studies show that this resilience is species dependent and can vary according to different plant characteristics and under different environmental conditions, including soil nutrient composition (e.g. nitrogen, zinc, copper and boron) and anthropogenic disturbance (e.g. plant harvesting) [37, 41, 52]. The results confirmed previous findings that indicated both *Amaranthus sp.* [38, 53] and *B. pilosa* [54, 55] have some level of drought tolerance and showed a significant difference in the species behavior in response to drought stress. Also, the resilience of both species differed depending on what plant characteristics were considered, although with some variations (especially for *Amaranthus sp.*) across the two treatment periods (i.e. after first and second harvests). Studies

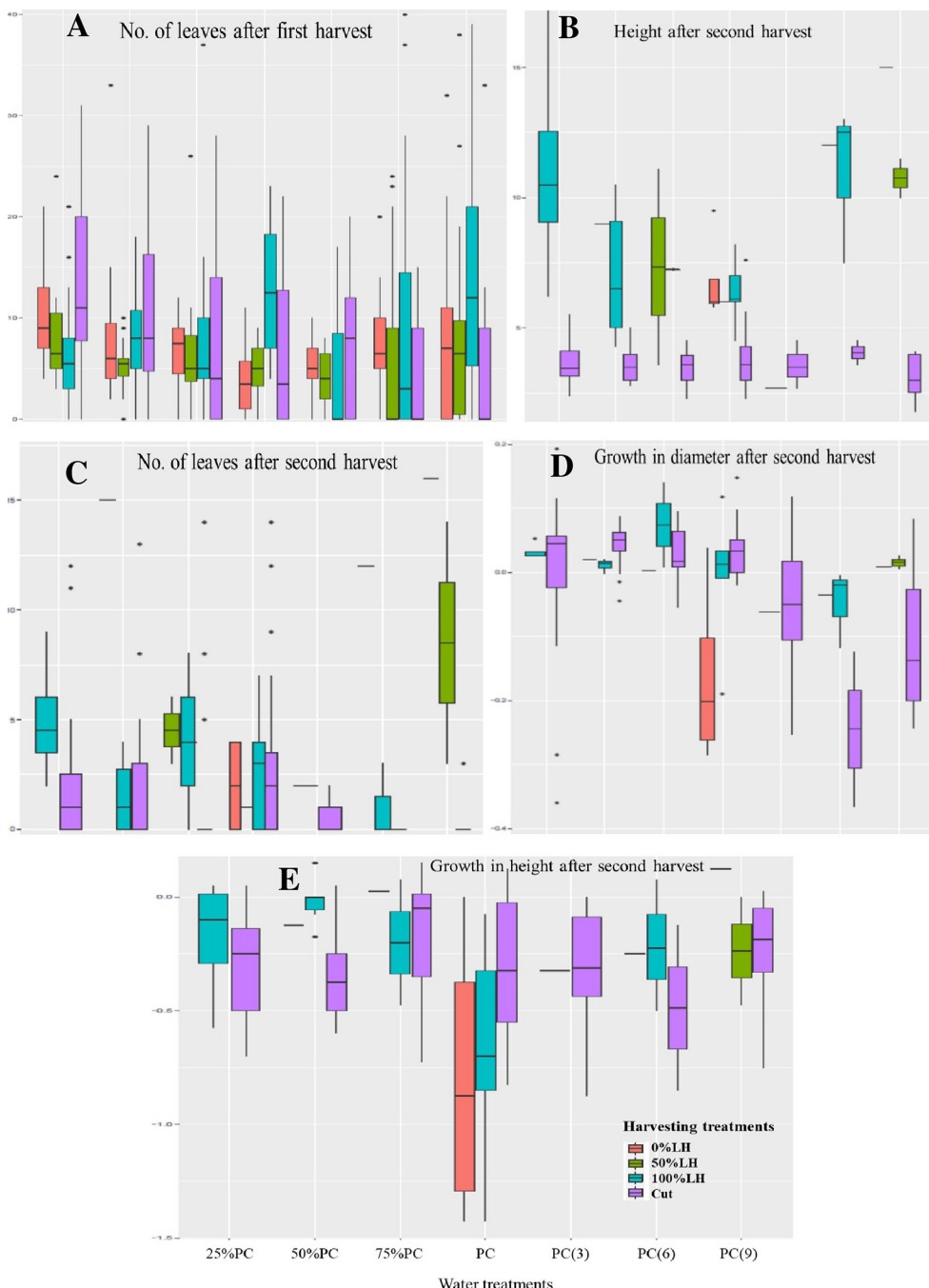

**Fig 2. Plant morphology (height), growth (in basal diameter and height) and leaf production (No. leaves per plant) of *Amaranthus sp*. according to water and harvesting treatments.** PC = pot capacity; treatments PC(3), PC(6) and PC(9) watered respectively every three, six, nine days, while the other treatments watered on daily basis.

on the impacts of both drought and harvesting, which are vitally important for the sustainable management, are lacking for *B. pilosa* in South Africa [56], while investigation on the concurrent impacts of these two drivers on plant performance are scarce in the literature.

The results suggest that *Amaranthus sp*. has only limited drought resiliency, while *B. pilosa* is the opposite. Previous studies have shown that although *Amaranthus* species can withstand

drought stress, an increase in the severity of drought stress reduced plant height and leaf yield [38, 57, 58]. In fact, *Amaranthus* plants can limit their height or leaf production in response to drought stress, as a drought avoidance strategy. Furthermore, our results indicate that *Amaranthus sp.* is least resilient to a decrease in the frequency of watering. In contrast, *B. pilosa* is drought tolerant under any level of drought stress. Previous studies elsewhere already emphasized the drought tolerance of *B. pilosa* under severe drought stress [54, 55]. According to previous studies, *Bidens* species possess a great ability to easily exploit the soil resources (i.e. water and nutrients), allowing them to compete with other species in harsh environmental conditions [59, 60]. According to De Freitas *et al*. [55], *Bidens* is able to reduce water loss through stomatal closure, hence lowering the transpiration rate. This mechanism allows *Bidens* to thrive under water scarce conditions. However, such mechanisms could also limit photosynthesis and lower the growth rate of individual plants [61]. This indicates that a condition of severe drought could have potential side-effects on *B. pilosa* performance, although the species can survive under harsh conditions. The results indicated that *B. pilosa* is more drought resilient than *Amaranthus sp*. under similar environmental conditions, echoing Pereira *et al*. [54] who showed *B. pilosa* to be better adapted to drought conditions than *Raphanus raphanistrum*.

This study highlights some variations in plant responses to drought stress between the two treatment periods (i.e. after first and second harvests), mainly for *Amaranthus sp*. Consequently, concerning the drought resiliency reflected in the plant characteristics, different trends were observed between the different treatment periods for *Amaranthus sp*., while for *B. pilosa*, trends were similar. Indeed, after the first harvest, in *Amaranthus sp*., height, growth in height, plant survival and leaf production appeared more drought-resilient than basal diameter and growth in diameter, while plant height and leaf production showed more drought resilience than plant survival, basal diameter and growth in diameter after the second harvest. Contrary to our results, reduction in plant height has previously been identified as a drought tolerance strategy in *Amaranthus* species [38, 58]. Also, Bangar *et al*. [62] showed the reduction in plant height was a mechanism to limit drought stress in different varieties of mungbean (*Vigna radiata* L.) in India. The differences between our results and previous findings may be explained by the concurrent impact of leaf harvesting in our study. Furthermore, the difference in plant responses to drought stress between the two treatment periods and the study species was also highlighted through the negative growth values that were observed mainly after the second harvest and in *Amaranthus sp*. These negative values could be due to the impact of prolonged drought causing thinning in plants which later start to dry and break before dying. In *B. pilosa*, plant survival and leaf production were more resilient than the other characteristics, after both the first and second harvests. Contrary to our results, De Freitas *et al*. [55] observed a reduction in the number of leaves per plant under water stress of 25% field capacity, as a mechanism to reduce water loss. However, the authors found no significant effect on leaf production. These differences in our results could be due to other influencing factors such as ecological conditions of the experiment. Although *B. pilosa* appears very drought tolerant, an increase in the severity of drought may affect either positively or negatively the composition of important nutrients and other compounds in the leaves, and reduce the quality of the seeds, hence of the seedbank. Sarker and Oba [63] highlighted an increase in the nutritional and bioactive compounds, phenolic acids, flavonoids and antioxidant capacity of the leaves of *Amaranthus tricolor* in Bangladesh due to drought stress. Also, the impact of drought stress on seed production has been highlighted in other leafy vegetables [64]. Further research is needed to provide more insight into the potential effects of severe and continuous drought on these aspects, for the wellbeing of the people using these species. Such study is also relevant in the case of *Amaranthus sp*. for the investigation of the impact of drought on nutritional properties of the leaves and young tender stems that are exploited for human consumption.

The impact of leaf harvesting differed between the two species; it was significant for *Amaranthus sp.* after both harvests, while it was not significant for *B. pilosa*, except for plant survival (after both harvests) and leaf production (after the first harvest). This corroborates previous research highlighting that the impact of harvesting plant parts is species specific depending on socio-ecological contexts [3]. Furthermore, for *Amaranthus sp.*, different trends were observed between the two treatment periods in terms of plant response to leaf harvesting, especially for leaf production. Indeed, after the first harvest, the more harvested plants displayed the highest performance in terms of basal diameter and its growth, leaf production and survival, while after the second harvest, the more harvested plants have less ability and resources remaining for them to maintain their stems and produce new leaves important for photosynthesis. This indicates a possible greater impact of prolonged high-rate leaf harvesting on at least the leaf production of *Amaranthus sp.* Concerning *B. pilosa*, the least harvested plants performed better than the most harvested ones for the considered characteristics (except leaf production after second harvest), although the differences were only significant for plant survival and number of leaves per plant (after first harvest). This suggests a potential, albeit weak, impact of leaf harvesting on this species meaning that a high rate of leaf harvesting can hinder the growth and production performance of *B. pilosa*. This also indicates the influence of other factors (e.g. drought stress, leaf attacks by insects and parasites) on the plant response to leaf harvesting, which needs to be further investigated.

## Effect size of individual impacts and interaction of drought and leaf harvesting

Overall, the results showed a significant interaction of drought and leaf harvesting on only *Amaranthus sp.* The interaction was significant on the combination of leaf production, plant survival, morphology and growth for *Amaranthus sp.* (after the second harvest, except for basal diameter), but not significant for *B. pilosa* after both the first and second harvests. However, in *Amaranthus sp.*, the interaction of the two drivers was significant on plant survival and leaf production after the first harvest. Also, the effect size of the interaction of in *Amaranthus sp.* was medium. This indicates that although the impact was significant, the variability between the groups of treatments that is really explained by the interaction of the two factors is medium [50]. In the same line, the impact of water treatment on the combination of the morphology, growth and leaf production was weaker than that of leaf harvesting for *Amaranthus sp.* In contrast, the individual impacts of both water treatment and leaf harvesting were weak in *B. pilosa*. This suggests that leaf harvesting has a greater effect on some TAVs used as NTFPs (e.g. *Amaranthus sp.*) than drought. Consequently, the impact of climate change on some TAVs might be less than that of harvesting. However, water is a necessary resource for plant growth and development and so the impact of severe drought (on even the drought tolerant species) might make plants more vulnerable to the impact of harvesting, at least for leaf production and plant survival. According to Gaoue *et al.* [13], drought stress can aggravate the impact of harvesting on plant species. This has been shown by this study through the significant interaction of the two factors, especially on the number of leaves per plant and plant survival for *Amaranthus sp.*

In general, this study highlights the negative impact of prolonged leaf harvesting on plant performance under the different levels of drought stress. However, according to the results, it seemed that some level of harvesting was beneficial. Furthermore, after the first harvest, the results showed that the plants that were cut survived better than the other plants under conditions of drought experienced in terms of reduction in the daily amount of water. Conversely, under drought situation of reduction in the frequency of watering (mainly after six and nine

days), plants with all or 50% leaves harvested survived better after the unharvested ones. This indicates that under severe drought, the removal of all the leaves together with the supporting stem is not a sustainable harvesting strategy in the case of *Amaranthus sp*. in comparison with the harvesting of the leaves alone. This is even more so in the context of prolonged harvesting as the results (Fig 2) showed a negative impact on plant performance (morphology, growth and leaf production), although the cut plants survived better. Thus, though the leaves can be harvested, some leaves (the small new ones) should be left on the plants to allow for continuation of photosynthesis. Finally, the results suggest that other ecological factors (such as soil nutrients) might have influenced the impact of the interaction of the two factors, and this need to be investigated.

## Limitations of the study

One limitation of this study might be that no fertilizer was added during the study, and thus it is unknown if any nutrient limitation might have influenced the results. Previous research has shown that plant response (e.g. morphology, growth and survival) to the impact of drought and harvesting may depend on the concentration of the main soil nutrients (e.g. nitrogen) [3, 41]. Also, the soil used was clayey, resulting in a degree of soil compaction. Soil compaction can influence the survival in some plants [65]. However, the different treatments were under the same soils and environmental conditions, and the above-mentioned potential impacts of non-use of fertilizer and soil compaction similarly applied to all the treatments. Also, clay soils are generally regarded to be rich in soil nutrients.

## Acknowledgments

Huge thanks go to Annegret Mostert for providing the *Amaranthus sp*. seeds, Rhodes Restoration Research group (within the Department of Environmental Science, Rhodes University) for allowing us to use their greenhouse and other materials for the experiment, and to Luvuyo Ncula and Siyamamisela Tinise for field assistance.

## Author Contributions

**Conceptualization:** Gisele K. Sinasson S., Charlie M. Shackleton.

**Formal analysis:** Gisele K. Sinasson S.

**Funding acquisition:** Charlie M. Shackleton.

**Investigation:** Gisele K. Sinasson S.

**Methodology:** Gisele K. Sinasson S.

**Supervision:** Charlie M. Shackleton.

**Writing – original draft:** Gisele K. Sinasson S.

**Writing – review & editing:** Charlie M. Shackleton.

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
