## [Decision Letter · Decision Letter 0]

4 Jan 2023

PONE-D-22-25341The concurrent impacts of drought and leaf harvesting on two traditional African vegetable non-timber forest product speciesPLOS ONE

Dear Dr. Sinasson S.,

Thank you for submitting your manuscript to PLOS ONE. After careful consideration, we feel that it has merit but does not fully meet PLOS ONE’s publication criteria as it currently stands. Therefore, we invite you to submit a revised version of the manuscript that addresses the points raised during the review process.

We look forward to receiving your revised manuscript.

Kind regards,

Cataldo Pulvento

Academic Editor

PLOS ONE

and https://journals.plos.org/plosone/s/file?id=ba62/PLOSOne_formatting_sample_title_authors_affiliations.pdf.

“All authors acknowledge funding support given by the South African Research Chairs Initiative of the Department of Science and Technology and the National Research Foundation of South Africa (grant no. 84379). Any opinion, finding, conclusion or recommendation expressed in this material is that of the authors. Huge thanks go to Annegret Mostert for providing the Amaranthus sp. seeds, and to Luvuyo Ncula and Siyamamisela Tinise for field assistance.”

“This work was supported by the South African Research Chairs Initiative of the Department of Science and Technology and the National Research Foundation of South Africa (grant no. 84379) hold by CMS (https://www.ru.ac.za/researchgateway/researchfoci/sarchi/). The funders had no role in study design, data collection and analysis, decision to publish, or preparation of the manuscript.”

“The authors declare that they have no known competing financial interest or personal relationships that could have appeared to influence the work reported in this paper.”

Additional Editor Comments:

Dear Authors

The manuscript "The concurrent impacts of drought and leaf harvesting on two traditional African vegetable non-timber forest product species" needs major revisions according to the reviewer comments before to be considered for publication on Plos One.

Reviewers' comments:

Reviewer's Responses to Questions

**Comments to the Author**

1. Is the manuscript technically sound, and do the data support the conclusions?

Reviewer #1: Yes

Reviewer #2: Yes

2. Has the statistical analysis been performed appropriately and rigorously? 

Reviewer #1: Yes

Reviewer #2: Yes

3. Have the authors made all data underlying the findings in their manuscript fully available?

Reviewer #1: Yes

Reviewer #2: Yes

4. Is the manuscript presented in an intelligible fashion and written in standard English?

Reviewer #1: Yes

Reviewer #2: Yes

5. Review Comments to the Author

Reviewer #1: The concurrent impacts of drought and leaf harvesting on two traditional African vegetable non-timber forest product species

The manuscript is well-written and can be of interest to readers of Plos One. The study aimed to investigate leaf production, morphological and growth in response to drought and leaf harvesting in Amaranthus and Bidens. The results showed significant effects of drought on both species. Amaranthus was more resilient to reduction in daily watering than reduction in the frequency, while Biden was resilient to all the drought stress treatments. In Amaranthus basal diameter, growth, leaf production and survival increased with harvesting after first harvest. After second harvest, there was decrease in plant height and leaf production. In Biden the impact was only significant on survival and leaf production (after first harvest). The results showed the possible negative impact of harvesting on these species.

My major concern about this manuscript is related to the experimental procedure. Information about the soil water tension is missing. Likewise, there is no information about leaf water content or leaf water potential

Specifics:

Methods: This section needs major improvement.

A topsoil was used in the experiment.

However, Information about the water tension of the soil under the drought treatment is missing. This information is important; otherwise it will be difficult repeating the experiment. I suggest adding this information, if available.

Also, I suggest describing soil texture, as this trait affect water holding capacity.

It is also missing the leaf water potential (or leaf water content) of plant subjected to water stress. If available, I suggest adding these data.

Climatic conditions:

I suggest adding a figure describing the climatic conditions (light, relative humidity, temperature, and potential evapotranspiration) within the greenhouse during the experimental period.

Replications

L169: Information about the number of replications per treatment is missing. I suggest adding it.

L141: … The maximum temperature in … ranged 18ºC-32ºC in

Suggestion: … the experiment, ranged from 18ºC to xx ºC in ..

(see others).

Results:

L245: effect size of the drought … was slightly weaker for Amaranthus sp. than for B. pilosa

" was slightly weaker " ?

I suggest adding the p values after " was slightly weaker "

(see others)

The n value in Tables 2:

Table 2: The sample size (n value) is missing. I suggest adding the n for each species. See others (e.g. Table 3, 5, 6).

Discussion:

On the negative growth (e.g.Table 3, 6).

I suggest commenting on this topic

Reviewer #2: Introduction

- Is well written with enough citation on the subject matter.

Methods

- In the methodology, I suggest the author should include some pictures in order to attract the reader.

- Tables should be re-drawn scientifically

- However, there are some issues I have raised (see them in the attached revised manuscript) that needs to be addressed by the author

6. PLOS authors have the option to publish the peer review history of their article (what does this mean?). If published, this will include your full peer review and any attached files.

Reviewer #1: No

Reviewer #2: No

---

## [Author Response · Author response to Decision Letter 0]

18 Feb 2023

Journal requirements

The authors made all the necessary editing work to ensure that their manuscript meets the journal style 

requirements.

2. Please remove any funding-related text from the manuscript and let us know how you would like to update your Funding Statement. Please include your amended statements within your cover letter; we will change the online submission form on your behalf.

The funding-related text was removed from the manuscript. The amended statement has been included in the new version of the cover letter.

3. Please complete your Competing Interests on the online submission form to state any Competing Interests. If you have no competing interests, please state ""The authors have declared that no competing interests exist."" This information should be included in your cover letter; we will change the online submission form on your behalf.

The information related to the Competing Interests has been included in the cover letter, as requested.

Change to the Data Availability statement has been provided in the cover letter. The authors, hereby, confirm that the data related to the manuscript are now fully accessible through the provided DOI link.

Reviewer 1

Methods

A topsoil was used in the experiment. However, Information about the water tension of the soil under the drought treatment is missing. This information is important; otherwise it will be difficult repeating the experiment. I suggest adding this information, if available. Also, I suggest describing soil texture, as this trait affect water holding capacity. It is also missing the leaf water potential (or leaf water content) of plant subjected to water stress. If available, I suggest adding these data.

The authors thank the reviewer for the important comments. However, we were unable to collect data related to the water tension of the soil nor the leaf water potential during the experiment and so those data are not available. 

As far as the soil texture is concerned, this information has been added to the methods section accordingly.

I suggest adding a figure describing the climatic conditions (light, relative humidity, temperature, and potential evapotranspiration) within the greenhouse during the experimental period.

The authors would have love to include such figure in the manuscript and for that some iButtons have been installed within the greenhouse during the experiment. However, due to some technical issues beyond our control, we were finally unable to obtain the climatic data.

L169: Information about the number of replications per treatment is missing. I suggest adding it.

The number of replications per treatment has been added for both species.

L141: … The maximum temperature in … ranged 18ºC-32ºC in

Suggestion: … the experiment, ranged from 18ºC to xx ºC in ..

(see others).

Change has been made in the sentences to consider the reviewer’s comment. 

Results

L245: effect size of the drought … was slightly weaker for Amaranthus sp. than for B. pilosa

" was slightly weaker " ?

I suggest adding the p values after " was slightly weaker "

(see others)

The n value in Tables 2:

Table 2: The sample size (n value) is missing. I suggest adding the n for each species. See others (e.g. Table 3, 5, 6).

The required information has been added to the manuscript.

Discussion

On the negative growth (e.g. Table 3, 6).

I suggest commenting on this topic

The negative values of the growth have been commented in the discussion accordingly.

Reviewer 2

In the methodology, I suggest the author should include some pictures in order to attract the reader.

Pictures illustrating different phases of the experiment have been included, as suggested by the reviewer.

Tables should be re-drawn scientifically

The authors thank the reviewer for the comment but we would like to note that the tables have been designed following the journal guidelines. However, some modifications have been made for the tables to look a bit better.

L480 Where is the figure?

According to the journal guidelines, figures should not be included in the main manuscript file. Each figure must be prepared and submitted as an individual file. Figure captions must be inserted in the text of the manuscript, immediately following the paragraph in which the figure is first cited.

All other suggestions, corrections and remarks by reviewer 2 for the improvement of the manuscript have been considered.

---

## [Decision Letter · Decision Letter 1]

20 Mar 2023

The concurrent impacts of drought and leaf harvesting on two traditional African vegetable non-timber forest product species

PONE-D-22-25341R1

Dear Dr. Sinasson S.,

We’re pleased to inform you that your manuscript has been judged scientifically suitable for publication and will be formally accepted for publication once it meets all outstanding technical requirements.

Kind regards,

Cataldo Pulvento

Academic Editor

PLOS ONE

Additional Editor Comments (optional):

Dear Authors

the manuscript "he concurrent impacts of drought and leaf harvesting on two traditional African vegetable non-timber forest product species" is accepted in the current form

Reviewers' comments:

Reviewer's Responses to Questions

**Comments to the Author**

1. If the authors have adequately addressed your comments raised in a previous round of review and you feel that this manuscript is now acceptable for publication, you may indicate that here to bypass the “Comments to the Author” section, enter your conflict of interest statement in the “Confidential to Editor” section, and submit your "Accept" recommendation.

Reviewer #1: All comments have been addressed

Reviewer #2: (No Response)

2. Is the manuscript technically sound, and do the data support the conclusions?

Reviewer #1: Yes

Reviewer #2: Yes

3. Has the statistical analysis been performed appropriately and rigorously? 

Reviewer #1: Yes

Reviewer #2: Yes

4. Have the authors made all data underlying the findings in their manuscript fully available?

Reviewer #1: Yes

Reviewer #2: Yes

5. Is the manuscript presented in an intelligible fashion and written in standard English?

Reviewer #1: Yes

Reviewer #2: Yes

6. Review Comments to the Author

Reviewer #1: Suggestions I made on the original version were observed, and those not taken into account were properly addressed.

I have no further comment our suggestion.

Reviewer #2: I have gone through the revised manuscript and realize that, authors have addressed all concerns that I

raised and I am now comfortable and the manuscript can be further processed for publication

7. PLOS authors have the option to publish the peer review history of their article (what does this mean?). If published, this will include your full peer review and any attached files.

Reviewer #1: **Yes: **Ricardo A. Marenco

Reviewer #2: **Yes: **Dr. Mhuji Kilonzo

---

## [Editor Report · Acceptance letter]

28 Mar 2023

PONE-D-22-25341R1 

The concurrent impacts of drought and leaf harvesting on two traditional African vegetable non-timber forest product species 

Dear Dr. Sinasson S.:

I'm pleased to inform you that your manuscript has been deemed suitable for publication in PLOS ONE. Congratulations! Your manuscript is now with our production department. 

Kind regards, 

on behalf of

Dr. Cataldo Pulvento 

Academic Editor

PLOS ONE